# A Concerted Action of UBA5 C-Terminal Unstructured Regions Is Important for Transfer of Activated UFM1 to UFC1

**DOI:** 10.3390/ijms22147390

**Published:** 2021-07-09

**Authors:** Nicole Wesch, Frank Löhr, Natalia Rogova, Volker Dötsch, Vladimir V. Rogov

**Affiliations:** 1Institute of Biophysical Chemistry, Center for Biomolecular Magnetic Resonance, Goethe-University Frankfurt, 60438 Frankfurt am Main, Germany; Wesch@bpc.uni-frankfurt.de (N.W.); Murph@bpc.uni-frankfurt.de (F.L.); Rogova@bpc.uni-frankfurt.de (N.R.); 2Structural Genomics Consortium, Buchmann Institute for Life Sciences, Goethe-University Frankfurt, 60438 Frankfurt am Main, Germany; 3Institute of Pharmaceutical Chemistry, Goethe-University Frankfurt, 60438 Frankfurt am Main, Germany

**Keywords:** UFM1, UBA5, UFC1, protein-protein interactions, NMR, complex structure

## Abstract

Ubiquitin fold modifier 1 (UFM1) is a member of the ubiquitin-like protein family. UFM1 undergoes a cascade of enzymatic reactions including activation by UBA5 (E1), transfer to UFC1 (E2) and selective conjugation to a number of target proteins via UFL1 (E3) enzymes. Despite the importance of ufmylation in a variety of cellular processes and its role in the pathogenicity of many human diseases, the molecular mechanisms of the ufmylation cascade remains unclear. In this study we focused on the biophysical and biochemical characterization of the interaction between UBA5 and UFC1. We explored the hypothesis that the unstructured C-terminal region of UBA5 serves as a regulatory region, controlling cellular localization of the elements of the ufmylation cascade and effective interaction between them. We found that the last 20 residues in UBA5 are pivotal for binding to UFC1 and can accelerate the transfer of UFM1 to UFC1. We solved the structure of a complex of UFC1 and a peptide spanning the last 20 residues of UBA5 by NMR spectroscopy. This structure in combination with additional NMR titration and isothermal titration calorimetry experiments revealed the mechanism of interaction and confirmed the importance of the C-terminal unstructured region in UBA5 for the ufmylation cascade.

## 1. Introduction

UFM1 is a small ubiquitin-like (UBL) protein spanning 85 residues. Like other UBLs, it has a low sequence identity to ubiquitin, but shares its specific (β-grasp) fold [1,2]. Unlike other UBLs (except for SUMO), UFM1 has a single C-terminal glycine residue, by which UFM1 gets attached to target proteins using an E1-E2-E3 enzymatic cascade [1,3,4]. Initially, the UFM1 precursor protein gets processed by the two specific proteases UfSP1 and UfSP2 to expose the C-terminal glycine residue [5,6,7]. Processed UFM1 gets activated by UBA5 (E1), a member of the ubiquitin-activating protein family [8,9,10], from which activated UFM1 is transferred to the catalytic cysteine 116 of UFC1 (E2) [1,8,11]. The last step is the transfer of UFM1 to the target proteins mediated by the specific UFM1 ligase 1 (UFL1), showing no typical E3 ligases domain organization [1,12]. The mechanism of this step is largely unknown and other proteins could be required for UFL1 ligase activity as well [13,14,15,16].

The first identified target of UFM1 was Ufm1-binding protein 1 (UFBP1, also known as DDRGK1 or C20orf116) [12]. Since then, discovery of new targets for UFM1 and the characterization of functional consequences of their ufmylation has constantly increased. Recently, new ufmylation targets involved in cancer progression [16,17], DNA damage response [18,19], translation machinery [20] and ribosome functioning [13,14] have been identified. Taking in account the broad range of biological pathways affected by ufmylation, it is not surprising that impaired ufmylation can be connected to many human diseases [16,21,22,23,24] and seems to be essential for embryonic development [25,26,27].

The exact mechanism of ufmylation and the full range of physiological consequences are not well investigated yet. The key elements of the ufmylation cascade (UBA5, UFC1, UFL1) show significant evolutionary differences to the well characterized enzymatic UBL cascades (e.g., ubiquitin or NEDD8) resulting in a number of structural and functional deviations from the canonical E1-E2-E3 pathways [3,4,28]. In contrast to other E1 family members, UBA5 does not display the characteristic domain architecture [28]. This 404-residue protein possesses a single well-folded adenylation domain (residues 57–329), comprising the active site Cys250 and provides a platform for ATP binding and UFM1 activation [8,29]. Two UBA5 regions—the N-terminal (1–56) and the C-terminal (334–404) segments—appear to be important regulatory elements for the function of UBA5 and in the ufmylation cascade. The N-terminal segment 1–56 (absent in one of the two existing UBA5 splice isoforms) significantly enhances ATP binding and therefore increases efficiency and velocity of UFM1 activation. Additionally, the N-terminal extension accelerates UFM1 transfer to UFC1 from the UBA5~UFM1 conjugate in presence of ATP [30].

The UBA5 C-terminal part (Figure 1A) plays a complex regulatory role, consisting of a few conserved regions that mediate interaction of UBA5 with other key players in the ufmylation cascade [31]. The first sequence is a conserved region (R1, residues 334–348), interacting with UFM1 [10,29,30,31,32] and also with LC3/GABARAP proteins [31,33]. This region (called LIR/UFIM by its dual nature) is important for the initial binding of UFM1 to UBA5 [10,29,31,32] and for the following UFM1 activation in a *trans*-fashion [29]. *Trans-*activation means that UBA5 forms an active homodimer, like other non-canonical E1 enzymes, and UFM1 bound to the LIR/UFIM segment of one monomer exposes its C-terminal Gly83 residue to the catalytic Cys250 of the other monomer [29]. GABARAP (and to a lesser extend LC3) proteins interact with the same UBA5 region and inhibit UFM1 binding to UBA5, thus modulating the conjugation of UFM1 to UBA5 and to UFC1 in vitro [31]. No evidence for the activation of LC3/GABARAP proteins by UBA5 was found so far. However, we showed previously that interaction between GABARAP proteins and UBA5 facilitates membrane localization of the latter [33].

The second region (R2, residues 364–372) is significantly less conserved among different species than the first region, with only Gly367 being evolutionary invariant. The role of this region is not understood, and no interacting proteins could be identified so far. However, a A371T mutation in the human protein located in this region decreases the ability of UBA5 to activate UFM1, to transfer the activated UFM1 to UFC1 and to mediate UFBP1~UFM1 formation [25,34].

Another conserved region in UBA5 is located at it very C-terminus (R3, residues 393–404) and is predicted to have a helical conformation. Initially, it was postulated by analogy with canonical E1 enzymes that the UBA5 C-terminal part possesses an ubiquitin-fold domain, mediating UBA5 interaction with UFC1 [8,11]. Later it was shown that a short UBA5 peptide (residues 381–404) is solely responsible for this interaction [32]. UFC1, the only known E2 enzyme for UFM1, was characterized structurally [11,35] a few years after discovery of the UFM1 cascade [1]. The common architecture of E2 enzymes—four α-helices, four β-strands and one 3_10_-helix (reviewed in [28])—is conserved for the UFC1 core (25–157). Lack of C-terminal α-helices and conserved motifs as well as the presence of an N-terminal α-helix, which stabilizes the UFC1 structure [11] result in structural differences, which classify UFC1 as a non-canonical E2 enzyme. Computational modeling (based on the existing crystal structure of the E1:E2 complex for the NEDD8 cascade) revealed that the second α-helix in UFC1 is the most probable site for interaction with UBA5. Indeed, the UFC1 K33A mutation significantly reduces both UBA5 binding and UFM1 transfer from UBA5 to UFC1 [11].

Despite these previous investigations, structural aspects and molecular mechanisms of the interaction between UBA5 and UFC1 are still largely unknown. Additionally, it is not clear, if other factors (e.g., UFM1 conjugated or bound to UBA5, or UFC1) could affect this interaction. In order to fill this gap, we systematically analyzed by isothermal titration calorimetry and NMR spectroscopy the interactions between different UBA5 fragments and UFC1, UFM1 and LC3/GABARAP proteins. Using this knowledge, we solved the solution structure of UFC1 in complex with an optimized C-terminal fragment of UBA5. Finally, our biochemical experiments showed the importance of the UBA5:UFC1 interaction for effective ufmylation.

## 2. Results

### 2.1. The UBA5 C-Terminal Part Is a Regulatory Platform for the Ufmylation Cascade

In order to understand the importance of the whole UBA5 C-terminal part and the roles of its individual conserved regions, we cloned and expressed a set of constructs containing the whole C-terminus, individual conserved regions and their combinations (Table 1) and investigated their interaction with the key elements of the ufmylation cascade.

First, we analyzed the effect of the UBA5 C-terminus on UFM1 transfer to UFC1 with an in vitro thioester formation assay (Figure 1B–E). Using UBA5 full length protein as E1 enzyme, we observed fast formation of a UFC1~UFM1 conjugate (~90% UFC1 was conjugated to UFM1 within 30 min, Figure 1B). When we used C-terminally truncated UBA5 (only the adenylation domain—AD, residues 1–330) as E1 enzyme, formation of a UFC1~UFM1 conjugate was significantly reduced (less than 5% UFC1~UFM1 conjugation was reached within 30 min; 7 h were needed to reach 80% UFC1~UFM1 conjugation, Figure 1C). However, transfer of UFM1 to UFC1 was rescued when we used an equimolar mixture of the UBA5 AD and the UBA5 C-terminal part as E1 enzyme. In this case, the ure 1D). These results indicate a crucial role of the UBA5 C-terminal part in the ufmylation cascade.

The most important regions in the UBA5 C-terminal parts—R1 (containing the LIR/UFIM sequence) and R3 (containing the UFC1 binding sequence)—seem to have a cumulative effect on the ability of UBA5 to transfer activated UFM1 on UFC1. Addition to the reaction mixture (UBA5 AD^1–330^, UFC1, UFM1, ATP/Mg2^+^) of UBA5 peptides lacking either the R1 or R3 sequences led to a reduced conjugation rate (Figure 1E and Appendix A). The results also indicate that the LIR/UFIM sequence is more important for the ufmylation cascade than the R3 site and that the conserved region R2 could also play an additive role in this process: the level of UFC1~UFM1 conjugates reached in reactions with AD^1–330^/R1-R2^325–376^ a higher level than when the R1^325–357^ peptide was added alone. Similarly, the addition of the isolated R2^359–376^ and R3^381–404W^ peptides had virtually no effect on the ufmylation reaction (Figure 1E and Appendix A).

UBA5 mutations within the R2 sequence (A371T and it phosphomimicking variant A371E) did not affect significantly the formation of the UFC1~UFM1 conjugate (Appendix A), indicating that the mutation becomes important for downstream events in the ufmylation cascade—potentially during binding of UBA5 to the membrane-associated E3 ligase (UFL1), to targets (UFBP1 [12], ASC1 [16], p53 [17], etc.) or important for other regulatory events. However, in another assay, using a mixture of wild type and mutated full length UBA5 proteins, we observed a small but reproducible reduction of UFC1~UFM1 conjugation (Appendix A).

Taken together we were able to restore the UFM1 transfer to UFC1 with separated AD and C-terminal peptides. With the single AD and only one of the regions the reaction took 7 h. The reaction rate increased by addition of peptides containing two regions and was similar to the full length UBA5 containing the complete C-terminal part.

### 2.2. Interactions between Different UBA5 C-Terminal Regions and UFC1, UFM1 and LC3/GABRAP Proteins

To understand how the UBA5 C-terminus participates in the ufmylation cascade, we performed isothermal titration calorimetry (ITC) experiments, in which we titrated UBA5 C-terminal peptides (see Table 1) to the UBA5 AD, UFC1, UFM1 and representative LC3/GABARAP proteins (Figure 2A, Appendix A and Table 2). The ITC experiments revealed that the entire UBA5 C-terminus (R1-R2-R3^325–404^) does not interact with the UBA5 AD, forming an independent UBA5 domain (Appendix A). The affinity of UFM1 to the R1-containing peptides (R1-R2-R3^325–404^ and R1-R2^325–376^, Appendix A and Table 2) does not change significantly compared to the affinity of the isolated R1^325−357^ peptide [31], indicating that this interaction is completely located within the LIR/UFIM containing region.

In contrast, LC3/GABARAP proteins showed a 10-fold higher affinity to the R1-R2-R3^325–404^ and R1-R2^325–376^ peptides compared to the isolated LIR/UFIM motif (R1^337−348^) characterized in [31,33]. The K_D_ values for interactions between R1-R2-R3^325–404^ and GABARAPL2 (0.17 µM) or LC3B (3.7 µM) indicate the same subfamily-specific preferences that were reported previously (Appendix A).

The affinity of the interaction between UBA5 and UFC1 has not been characterized previously. In ITC experiments, the shortest UBA5 peptide spanning the R3 sequence (R3^388–404^) bound to UFC1 with a K_D_ of >11 µM. The affinity increased 3-fold for R2-R3^359–404^ and R1-R2-R3^325–404^ peptides (K_D_ of 2.7 and 2.4 µM, respectively; Figure 2A and Table 2).

UFM1 and LC3/GABARAP proteins did not show interaction with the R2 region. However, R1-R2-R3^325–404^ peptides containing A371T and A371E mutations showed some increase in affinity to LC3B and GABARAPL2 proteins but not to UFM1 and UFC1 (Appendix A, Table 2).

To understand the role of the UBA5 C-terminal region in coordination of the binding events reported above on the molecular level, we performed NMR titration experiments. In those experiments, we titrated non-labeled UFC1 and GABARAPL2 proteins to a ^15^N-labeled R1-R2-R3^325–404^ peptide. The NMR experiments revealed that the interaction between UFC1 and UBA5 is mediated mostly by the UBA5 residues 386–404. These residues (in contrast to the vast majority of the R1-R2-R3^325–404^ resonances, which are not affected by addition of UFC1) showed a slow-to-intermediate exchange mode. The amide backbone resonances of these residues disappeared with small chemical shift perturbation (CSP) at the earlier stages of titrations and did not appear again up to an 8-fold molar excess of UFC1 (Figure 2B, the full size spectra are presented in Appendix A). UBA5 residues 383–386, 400 and 403 appeared to be in intermediate exchange mode (their amide backbone resonances displayed CSP with intensity change, however, they became visible at the latest titration steps). It seems, that these UBA5 residues form additional interactions with UFC1. Interestingly, a subset of the residues within the R2 region (V370, A371, Y372 and T373) displayed moderate CSPs, however, below standard deviation level (Figure 2C), possibly indicating an influence of the UBA5 A371T mutation on the recognition of UFC1.

The GABARAPL2 titration to the R1-R2-R3^325–404^ peptide revealed a complex behavior of interactions between these two polypeptides (Appendix A). At the earlier stages of titrations (until a molar ratio of 1:1) the R1-R2-R3^325–404^ resonances showed significant CSPs (in slow-to-intermediate exchange mode), mostly within the LIR/UFIM region (residues D338-V349). Moderate CSPs (with magnitudes above one standard deviation level) can also be observed in sequences adjacent to the R1 peptide: I335 N-terminally, and E352-S358 C-terminally. However, increased concentrations of GABARAPL2 induce further CSPs over the entire R1-R2-R3^325–404^ peptide sequence, including residues in R2 (A371-I374) and R3 (V382-G391, L394, D396, M398) regions. For the resonances within the R1 and adjacent sequences, the direction of the CSPs changed (Appendix A), while residues in R2/R3 regions approach the slow-exchange regime with increased CSP values. These observations indicate, that GABARAPL2 binds first to the LIR/UFIM region, and after saturation of this binding site, GABARAPL2 interacts with additional sites within the UBA5 C-terminus. According to this model, high concentration of GABARAPL2 could efficiently prohibit the UFC1~UFM1 conjugation, which was observed in ufmylation assays (Appendix A).

We could not observe any interactions between UFM1 and UFC1 proteins (using NMR titration of ^15^N-labeled UFC1 with non-labeled UFM1 up to 1:2 molar ratio). Additionally, binding of UFC1 to the R3 region within the UBA5 C-terminus^325–404^ did not initiate UFC1:UFM1 interactions as displayed by NMR experiments of ^15^N-labeled UFC1 in complex with the R1-R2-R3^325–404^ peptide titrated with non-labeled UFM1 until a 1:4 molar ratio. Furthermore, no interaction of ubiquitin to the UBA5 C-terminal region was observed, suggesting that the UBA5 C-terminus is specific for UFM1.

Taken together, we identified a UFC1-interacting region within the UBA5 C-terminus using ITC and NMR experiments. The region is slightly longer than the conserved R3 sequence which was detected previously and shows a micromolar affinity to UFC1. While UFM1 seems to bind only to the LIR/UFIM region of UBA5, LC3/GABARAP proteins interact with additional residues outside of the of the R1 sequence. LC3 and GABARAP subfamily proteins showed a 10-fold higher affinity to the complete UBA5 C-terminus compared to the isolated R1 peptide. Additionally, UFC1 showed interaction outside of the R3 region, binding residues within the R2 region. NMR titrations revealed that UFC1 and GABARAPL2 have a more complex binding mechanism to the UBA5 C-terminus, involving some residues in the R2 region. However, no direct interactions of all tested proteins to the isolated R2 peptide were observed.

### 2.3. Structure of UFC1 in Complex with the UBA5 R3 Peptide

To understand the interaction between UFC1 and UBA5 on a molecular level, we solved the NMR solution structure of UFC1 in complex with the UBA5 R3^381–404W^ peptide. Based on the results of our ITC and NMR experiments, we optimized the R3 peptide sequence including residues 381–404 of UBA5 and an additional C-terminal tryptophan residue (at position 405), providing a possibility to calculate the peptide concentration by UV spectroscopy. The R3^381–404W^ peptide displayed the expected ability to form a stable complex with UFC1. In contrast to the shorter R3^388–404^ peptide or to the R1-R2-R3^325–404^ peptide, the R3^381–404W^ peptide showed re-appearance of all resonances at the latest titration steps with UFC1 (Figure 2D and Appendix A). Correspondingly, almost all backbone amide resonances of UFC1 became visible at the latest stages of titration with R3^381–404W^ (Figure 2E and Appendix A), enabling us to solve the UFC1:R3^381–404W^ complex structure. The structure is presented in Figure 3 and Appendix A, structural statistics are given in Appendix A. The UFC1 structure in complex with the R3^381–404W^ peptide is close to the previously published X-ray and NMR structures of free UFC1 (Appendix A, [11,35]). The most significant differences were observed in the orientation of the N-terminal α-helix α1 (residues 1–11), the conformation of the C-terminal UFC1 part (residues 156–167) and the flexible loop near the active-cite cysteine 116 (residues 91–124, Appendix A).

Residues 394–404 of the R3 region form the predicted [32] α-helix, residues 384–392 are in an extended conformation, well-defined and occupy a specific area on the UFC1 surface. Residues 381–383 seem disordered and do not interact specifically with any UFC1 residues. The amphiphilic R3 α-helix is aligned to the α2 α-helix of UFC1 (Figure 3A) on the side opposite to the catalytic cysteine (C116). The UFC1 resonances on the C116 side were not affected upon NMR titration experiments, leading to the suggestion that this side could interact with the adenylation domain during UFM1 transfer. Sidechains of the R3^381–404W^ hydrophobic residues (L394, L397, M401 and M404) are placed into the large hydrophobic cleft formed by α-helix α2 and β-strand β1 of UFC1 (residues W28, V29, L32, Y36, L39, I40, V43, L56 and aliphatic moieties of K33 and Q37; Figure 3C). Two additional hydrophobic patches I and II (formed by residues within α-helices α1, α2 and the loop between them) accommodate UBA5 residues L385 and V386 (Figure 3D).

In addition to intermolecular hydrophobic interactions, the complex between UFC1 and the R3^381–404W^ peptide is stabilized by a network of intermolecular hydrogen bonds and polar contacts (Figure 3E, all intermolecular contacts detected by the LigPlot software for the UFC1:R3^381–404W^ complex are shown in Appendix A). The network covers almost all residues within the R3 region, which interact with the polar residues of UFC1 in the same area—α1, α2, loop between them and β-strand β1 (detailed information on the polar contacts is given in the Appendix A). The only additional UFC1 residue that forms intermolecular hydrogen bonds to the R3^381–404W^ peptide outside of this UFC1 region, is K131, whose sidechain is in close proximity to the carboxyl group of UBA5 E384.

Previously, it was predicted that the UFC1:UBA5 interaction is mediated by the UFC1 α-helix α2 [11] and the point mutation K33A within this helix impaired UBA5 binding and UFM1 transfer to UFC1, whereas Q31A had no effect. In our structure we observed that the UFC1 K33 sidechain forms an intermolecular hydrogen bond with the UBA5 D389 sidechain (Figure 3F). In contrast, UFC1 Q31 is not in contact with any of the UBA5 R3 residue and could not affect the UBA5:UFC1 interaction. 

In summary, the structure of UFC1 in complex with the R3^381–404W^ peptide revealed that the C-terminal α-helical part of UBA5 is pivotal for the attraction of UFC1 to UBA5. In addition to the α-helical part, UBA5 residues L385 and V387 also play a role in the UBA5 interaction with UFC1. The UFC1 hydrophobic groove and hydrophobic patches I and II are the most important areas mediating the interaction. Intermolecular polar contacts and hydrogen bonds stabilize the observed complex. The sidechain of UFC1 K33 is involved in an intermolecular hydrogen bond formation (to UBA5 D389 as a counterpart), therefore, its substitution to alanine interferes with the UFC1 interaction to UBA5 [11].

### 2.4. Interactions within the Ufmylation Cascade

Our results so far describe the interaction of UFC1 with the UBA5 C-terminal region. However, the interaction between full length UBA5 and UFC1 could be more complex and could depend on UFM1 conjugation to UBA5 or UFC1. To answer the question if UBA5 can bind UFC1 via additional sites, we analyzed NMR spectra of UFC1 with a 2-fold excess of unlabeled UBA5 FL^1–404^. We did not observe significant CSPs (shift or disappearance of the UFC1 resonances) in comparison to the spectra of the UFC1:R3^381–404W^ complex (Appendix A).

Additionally, UBA5 lacking the R3 region (ΔR3^1–380^) did not interact with UFC1 (as observed by NMR titration experiment, Appendix A) and significantly slowed down UFM1 transfer to UFC1 (Figure 4A, Appendix A). All these observations indicate that besides R3, UFC1 does not bind to any UBA5 regions efficiently. However, even weak additional interactions could facilitate the UFC1~UFM1 conjugation as observed in this work for the UBA5 constructs lacking R3 (Figure 1C and Figure 4A, Appendix A).

To investigate if conjugation of UFM1 to the UBA5 catalytic cysteine (C250) affects the UFC1:UBA5 interactions, we prepared full length UBA5 C250K mutant and stably conjugated UFM1 to it as reported before for a number of ubiquitin-specific E2 enzymes [38,39,40]. We compared the UFC1 spectra after addition of a twofold molar excess of FL^1–404^ and FL^1–404^ C250K~UFM1 constructs (Appendix A). Again, no significant enhancement of the UBA5:UFC1 interaction induced by the UBA5~UFM1 conjugation was observed. ITC experiments, in which we titrated UFC1 to FL^1–404^ and to FL^1–404^ C250K~UFM1 samples (Figure 4B, Table 2), showed small increases in their affinity to UFC1 in comparison to the R1-R2-R3^325–404^ peptide (K_D_ values for R1-R2-R3^325–404^, FL^1–404^, FL^1–404^ C250K~UFM1 are 2.4, 1.4 and 1.2 µM, respectively).

UFM1 conjugation to UBA5 C250 did not prohibit UFM1 binding to the R1 region. The gel-filtration profile and following electrophoretic analysis of the fractions showed that the FL^1–404^C250K~UFM1 but not the AD^1–330^ C250K~UFM1 peak contains non-conjugated UFM1 (Figure 4C and Appendix A).

## 3. Discussion

In this paper we analyzed the interactions between UBA5 and UFC1 enzymes within the ufmylation cascade and found that the unstructured UBA5 C-terminal part provides a platform for multiple protein–protein interactions affecting the efficiency of the activated UFM1 transfer from UBA5 to UFC1.

### 3.1. The UFC1:UBA5 Interaction

Our ITC and NMR titration experiments revealed that the interaction between UFC1 and UBA5 is mediated mostly by the relatively short and evolutionary conserved stretch of UBA5 residues (383–404). Using the optimized UBA5 construct (R3^381–404W^ peptide), we solved the NMR structure of the UFC1:R3 complex. The complex structure in combination with the NMR and ITC titration experiments revealed that in addition to the core R3 region, residues in the region R2 contribute to the interaction. While the isolated R2 peptide does not interact with UFC1, the combination of R2 and R3 binds three times tighter than the R3 alone. This weak additional interaction also explains the results of the UFC1 ufmylation assay (Figure 1). Ability of the isolated UBA5 AD to transfer activated UFM1 on UFC1 gets rescued by addition of the R1-R2-R3 peptide. In this peptide the R1 sequence can bind to UFM1 conjugated to UBA5 and recruit via its exposed R3 peptide UFC1 to the complex (Figure 4D). In full length UBA5 this recruitment occurs similarly, resulting in very similar UFC1 ufmylation rates. Adding only the R2-R3 peptide to the UBA5 AD increases the reaction rate only slightly above the isolated individual R1, R2 or R3 peptides, because deletion of the R1 sequence prevents effective recruitment of UBA5 C-terminus in complex with UFC1 to the UFM1-charged AD. A stronger rescue effect is seen for the R1-R2 peptide, because the R2 peptide probably still can interact with UFC1 (Figure 2C) and thus increase the local concentration of UFC1 around the AD. In the full length UBA5 protein, this recruiting effect most likely occurs *in-trans* [29]. A dimer was found in the crystal structure of UBA5 in complex with UFM1 bound to the R1 region. The linker between the AD and the R1 sequence is too short for an *in-cis* transfer to the active site cysteine, but within the dimer UFM1 bound to R1 of one monomer can be adenylated by the other UBA5 molecule of the dimer. This mechanism was confirmed by clever mutational engineering showing that a forced monomer cannot activate UFM1. Similarly, a trans mechanism was proposed for the transfer to UFC1 as well (Figure 4D). In our NMR titration experiments the UFC1 catalytic cysteine C116 and neighboring residues were not affected upon titration with the R3 peptide and our complex structure revealed that the R3 peptide occupies the side of the UFC1 molecule opposite to C116, indicating that the UFC1 surface around C116 could be used by the UBA5 AD during UFM1 transfer. Note that our data alone did not exclude *in-cis* UFM1 transfer mode.

In general, we were able to observe relatively stable interactions between members of the ufmylation cascade only for the R1:UFM1 and R3:UFC1 interactions. All other interactions are so weak that they are hard to detect by NMR (additional R2 residues with UFC1) or cannot be characterized at all. This includes interaction of UFC1 with the UBA5 AD alone or charged with UFM1 as well as with isolated UFM1. These results suggest that transfer of UFM1 from the adenylation domain of UBA5 to UFC1 uses in addition to relatively strong interactions for recruitment of the necessary components very weak interactions for the transfer (hit-and-run model).

### 3.2. Interaction between GABARAPL2 and UBA5 C-Terminal Part

The GABARAP and LC3 subfamilies members were found to bind UBA5 via an atypical LIR (LIR/UFIM), an evolutionary conserved sequence within the UBA5 C-terminal part [31,33]. The ITC and NMR experiments revealed additional interactions next to the known binding site within the R1. UBA5 constructs including both R1 and R2 regions showed a 10fold higher binding affinity to all GABARAP and LC3 protein subfamily members. Binding preference towards the GABARAP subfamily proteins remains preserved [31,33]. NMR titration experiments disclosed a more complex binding mechanism of GABARAPL2 to the complete C-terminal UBA5 peptide. At earlier titration steps, UBA5 residues within R1 were strongly affected by GABARAPL2 binding. However, with increasing concentrations of GABARAPL2 conserved residues located mostly in R2 started to display significant CSPs as well. These additional interactions might become relevant when UBA5 gets recruited to a membrane and GABARAP proteins cluster in micro-domains. A high concentration of GABARAP proteins in combination with a reduction of the search space for interactions from three to two dimensions could allow simultaneous binding of several GABARAP proteins to the UBA5 C-terminus. Recruitment of UBA5 to the membrane of the endoplasmic reticulum (ER) has been observed [33], the exact role of this recruitment is subject for further investigations.

### 3.3. The Role of the A371T Mutation in the Ufmylation Cascade

Many diseases are associated with impaired ufmylation [16,21,22,23,24]. Ufmylation is essential for embryonic development [25,26,27]. The A371T mutation was described previously to be present in patients suffering from severe infantile-onset encephalopathy [25,34]. Further investigations showed slightly reduced UBA5 thioester conjugation with UFM1 and reduced enzymatic activity in trans-thioesterification of UFC1 in vivo for the UBA5 A371T mutant [25,34]. Our ITC experiments with C-terminal UBA5 peptides containing the A371T or its phosphomimicking A371E mutations (located in the R2 region) showed almost no influence on UFM1:UFC1 binding affinity. NMR titration of the wild type ^15^N-labeled R1-R2-R3^325–404^ peptide with UFC1 displayed some moderate CSPs for the A371 and residues around, indicating a minor role of the R2 sequence in UFC1 binding. In vitro ufmylation assays showed that R1-R2-R3^325–404^ A371T and R1-R2-R3^325–404^ A371E peptides have nearly the same trans-thioesterification efficiency compared to wild type R1-R2-R3^325–404^ peptide in standard ufmylation assay conditions. However, reduction of ATP (to 25 µM) led to a reduction of the UFC1~UFM1 conjugate fraction for both mutated UBA5 peptides in comparison to wild type peptide, as reported previously [25,34].

Interestingly, we detected an increased affinity of R1-R2-R3^325–404^ A371T and R1-R2-R3^325–404^ A371E peptides to GABARAPL2 and LC3B proteins in ITC experiments. While GABARAPL2 showed a ~3-fold increased affinity to both mutated peptides in comparison to the wild type peptide, we detected a ~7-fold increased affinity for LC3B to the A371E mutant and a ~3-fold increased affinity to the A371T mutant. NMR titration experiments with wild type R1-R2-R3^325–404^ peptide revealed that A371 and adjacent residues are involved in GABARAPL2 binding at high GABARAPL2 concentrations. Again, taking into account that GABARAP and LC3 protein family members are proposed to recruit UBA5 to the ER membrane and play a critical role in the regulation of the ufmylation pathway [33,41], these results lead to the assumption that the A371T mutation plays a minor role in the ufmylation reaction itself, but might affect UBA5 localization and thus influences target ufmylation.

## 4. Materials and Methods

### 4.1. DNA Constructs Used in This Study

Genes of proteins and UBA5 peptides were cloned into a pET39_Ub19 vector containing a modified ubiquitin tag [33] and a TEV cleavage site resulting in a N-terminal cloning artefact of the first three residues (GAM). UBA5 C250K and UFC1_His6 were cloned into pNiC-CTH0 vector with a C-terminal hexahistidine-tag cleavable by an introduced TEV cleavage site. For site-directed mutagenesis PfuUltra II fusion HS DNA polymerase (Agilent Technologies Germany, Frankfurt, Germany) was used according to the manufacturer’s instructions. A comprehensive list of DNA constructs used in this study is given in Table 1.

### 4.2. Expression, Isolation and Purification of the Peptides and Proteins

All proteins and peptides were expressed in E.Coli T7 Express (New England Biolabs GmbH, Frankfurt, Germany) cells in LB or M9 (to obtain ^15^N- and ^13^C,^15^N-labeled polypeptides) media according to the protocol described in [33,36]. For protein purification, bacterial cell pellets were resuspended in lysis buffer (50 mM Tris-HCl pH = 7.5, 100 mM NaCl, 5% glycerol, 5 mM PIC (protease inhibitor cocktail)) and lysed via sonication (TT13 Sonotrode, 40% amplitude, for 6 × 1 min with a 0.5/0.5-s pulse). Lysates were centrifuged for 45 min at 17,000× *g* at 4 °C. Supernatants were loaded onto a His Trap Fast Flow 5 mL column (GE Healthcare, München, Germany) equilibrated in loading buffer (50 mM Tris-HCl pH = 8.0, 250 mM NaCl, 1% glycerol, 20 mM imidazole). The column was washed with loading buffer for 5–10 CV and protein was eluted with elution buffer (50 mM Tris-HCl pH = 8, 250 mM NaCl, 1% glycerol, 400 mM imidazole). Simultaneous TEV cleavage (1 mg TEV protease was added to 100 mg peptides/proteins) and buffer exchange to loading buffer via dialysis was performed over night at 4 °C. After reverse IMAC, proteins were concentrated with conical concentrators (Millipore Merck, Darmstadt, Germany) and loaded on a Superdex 10/60 75 or 200 column (GE Healthcare, München, Germany) for further purification and equilibration with ITC/NMR buffer (25 mM HEPES pH = 7.5, 100 mM NaCl). For structural NMR spectroscopy, buffer containing 50 mM Tris-HCl pH = 7.5, 100 mM NaCl was used. Prior to NMR experiments, TCEP and protease inhibitors cocktail were added to the samples to final concentrations 1 and 5 mM, respectively. Purified peptides and protein were concentrated and stored at −80 °C. The protein and peptide concentrations were calculated from the UV absorption at 280 nm by Nanodrop spectrophotometer (Thermo Scientific, Langenselbold, Germany).

### 4.3. In Vitro Thioester Formation Assay

Ufmylation reaction assays were adopted from work of Xie [32]. Briefly, 70 µM UFM1, 20 µM UFC1 and 20 µM of different UBA5 constructs were mixed in reaction buffer (50 mM HEPES pH = 7.5, 100 mM NaCl, 5 mM MgCl_2_). After starting the reaction with addition of 1 mM ATP, the reaction mix was incubated at 22 °C for the desired time. To quench the reaction and prepare electrophoretic samples, 1 µL of the reaction mix was added to 99 µL 1x non-reducing SDS loading buffer and frozen in liquid nitrogen. Sample content was visualized by sodium dodecyl sulfate polyacrylamide gel electrophoresis (SDS-PAGE). The transfer to polyvinylidene difluoride (PVDF) membrane was performed via a Trans-Blot^®^ Turbo™ Transfer System (Bio-Rad, München, Germany). After transfer the membrane was blocked with TBST (Tris-buffered saline with Tween20 buffer, 20 mM Tris, 150 mM NaCl and 0.1% TWEEN 20) containing 5% *w*/*v* nonfat dry milk for 1 h, followed by α-UFC1 antibody incubation over night at 4 °C (ab189251 abcam, 1:10,000 in TBST containing 5% *w*/*v* nonfat dry milk). After washing with TBST the membrane was incubated with secondary antibody (anti-rabbit-HRP) for 1 h at RT and again washed with TBST. Detection was performed by addition of ECL solution. For quantification of UFC1 ufmylation coloc2 software implemented in ImageJ was used. To show the kinetic differences between FL^1–404^ and ΔR3^1–380^ on UFC1 ufmylation, the reactions were started with 25 µM ATP.

For stable UBA5~UFM1 conjugation, 70 µM UFM1, 20 µM FL^1–404^ C250K and 1 mM ATP were added to ufmylation reaction buffer (50 mM HEPES pH = 10.0, 100 mM NaCl, 5 mM MgCl_2_). For NMR analysis, resulting complexes were concentrated and equilibrated with ITC/NMR buffer. To analyze complex formation by ufmylation assay 300 µL of sample were loaded onto a Superdex 200 10/300 column (GE Healthcare, München, Germany).

### 4.4. Isothermal Titration Calorimetry 

All ITC experiments were performed at 25 °C using a VP-ITC microcalorimeter (Malvern Panalytical Ltd., Malvern, UK). Peptides in concentration of ~400 µM were titrated into 20–25 µM solutions of corresponding binding partner at a stirring speed of 307 rpm. The raw data were corrected on the dilution heat of peptides obtained in independent experiment (titration of the peptide in syringe into the ITC/NMR buffer in the measuring cell). Pre-titration delay was set to 180 s, interval between titration steps was experimentally adjusted to avoid kinetic contribution to the observed heat effects and set to 200 s. A single ITC profile was collected for each type of interaction. The ITC data were analyzed based on a “one-site” binding model with MicroCal ITC software implemented in Origin 7.0.

### 4.5. NMR Spectroscopy

All NMR experiments were performed at a sample temperature of 25 °C on Bruker 600, 700, 800, 900, and 950 MHz spectrometers equipped with cryogenic probes, and a 500 MHz spectrometer equipped with a room-temperature triple-resonance probe. All NMR spectra were analyzed with the Sparky 3.114 software (University of California, San Francisco, USA). For NMR titration experiments, the non-labeled UBA5 peptides were titrated to 100 µM ^13^C,^15^N-labeled UFC1 to a final molar ratio of 1:8 (UFC1:UBA5 peptide). Conversely, 100 µM ^13^C,^15^N-labeled UBA5 peptides were titrated with non-labeled UFC1 to a final molar ratio 1:4 (UBA5 peptide:UFC1). 2D ^1^H-^15^N correlation spectra ([^15^N,^1^H] TROSY-HSQC) were recorded at each titration point. The same types of spectra were recorded to estimate binding of ^13^C,^15^N-labeled UFC1 (75 µM) to non-labeled UBA5, UBA5~UFM1 and UFM1 at 1:2 molar ratios. CSP values, Δδ, were calculated for each individual amide group using the formula Δδ = [(Δδ_N_/5)^2^ + Δδ^2^_HN_)]^1/2^.

For structural NMR spectroscopy, samples containing 1 mM ^13^C,^15^N-labeled UFC1 in the presence of 1 mM non-labeled R3^381–404W^ and 0.3 mM ^13^C,^15^N-labeled R3^381–404W^ in presence of 1.2 mM non-labeled UFC1 were used. As buffer condition 50 mM Tris pH = 7.5, 100 mM NaCl, 2 mM TCEP, 5 mM PIC, 5% D2O, 0.15 mM DSS was chosen. Backbone resonance assignment was performed using 3D BEST-TROSY versions [42,43] of HNCACB, HNCO, HN(CO)CACB and HN(CA)CO pulse sequences. Aliphatic ^1^H and ^13^C side-chain assignments resulted from (H)CC(CO)NH-TOCSY, and H(CCCO)NH-TOCSY experiments [44,45]. The assignment of aromatic side chain resonances was accomplished with amino-acid type specific versions of the (H)CB(CGCC-TOCSY)H^ar^ experiment [46] in conjunction with a [^13^C,^1^H]-ct-TROSY experiment [47,48] and an aromatic ^13^C-resolved 3D NOESY-SOFAST-HMQC experiment was used for verification. To obtain distance restrains for structure calculations 3D ^15^N- and ^13^C- separated NOESY-HSQC spectra, recorded with a mixing time of 60 ms, were analyzed. To obtain intermolecular distance restrains, 3D F1-^13^C/^15^N-filtered NOESY-[^13^C_ali_,^1^H]-HSQC, NOESY-[^13^C_aro_,^1^H]-SOFAST-HMQC and NOESY-[^15^N,^1^H]-SOFAST-HMQC experiments (mixing time 150 ms) were performed [49]. The structure was calculated via CYANA [50] version 3.98 with automated peak assignment. Torsion angles were predicted based on chemical shift values by PREDITOR program [51]. Restrained energy refinement using OPALp [52] was performed for the 20 conformers with the lowest final CYANA target function.

The 20 energy-refined conformers were deposited in the Protein Data Bank with accession code 7OVC. The chemical shift assignments were deposited in the BioMagResBank (BMRB) database with accession code 34638.

## Figures and Tables

**Figure 1 ijms-22-07390-f001:**
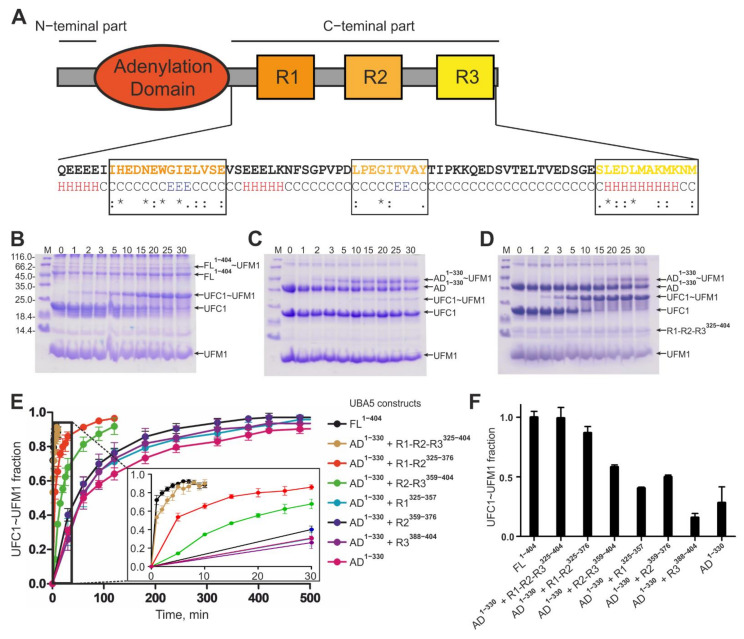
Role of C-terminal UBA5 regions on UFC1~UFM1 conjugation. (**A**) Overview of UBA5 conserved regions. Structure prediction (JPRED) and residue conservation are indicated below the C-terminal sequence (* indicates fully conserved residues; : indicates residues of high similarity; . indicates residues of low similarity). The different UBA5 C-terminal conserved regions are highlighted. (**B**–**D**) Gel electrophoresis of ufmylation assays including UBA5 FL^1–404^ (**B**), AD^1–330^ (**C**) and a mixture of UBA5 AD^1–330^ and R1-R2-R3^325–404^ (**D**) as E1 enzymes. Ufmylation was tracked over 30 min. Corresponding protein bands are labeled on the right side. (**E**) Ufmylation assays tracked over time with different UBA5 constructs indicated on the right side. The time points of 0–30 min are magnified. All assays were done as triplicates. Evaluation of UFC1~UFM1 conjugate was done via Western blotting. (**F**) Ufmylation assays quantified after 30 min reaction time. The fractions of the UFC1~UFM1 species are presented as bar diagram for each reaction mixture. For quantification of conjugated and unconjugated UFC1 coloc2 software implemented in ImageJ was used.

**Figure 2 ijms-22-07390-f002:**
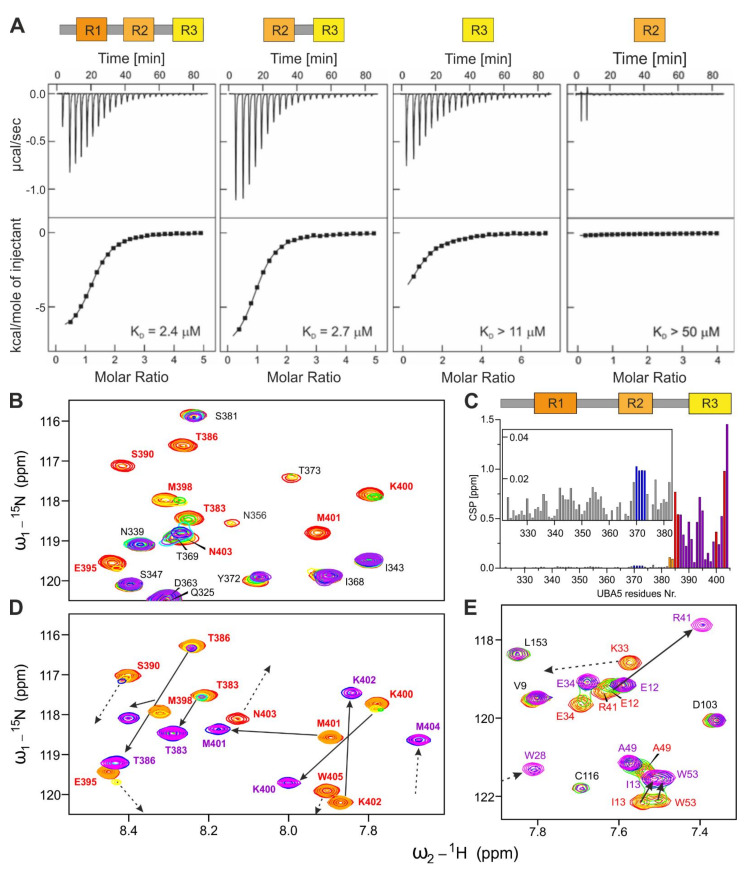
Interaction between UBA5 C-terminal fragments and UFC1 protein. (**A**) UFC1 binding to different UBA5 C-terminal peptides observed by ITC experiments. The upper graphs display the raw heat data; the lower graphs show the integrated heat per titration steps (black squares) with best-fit curve (line). The used peptides are graphically visualized above the corresponding titration profiles. K_D_ values are indicated. (**B**) NMR titration of ^15^N-labeled R1-R2-R3^325–404^ peptide with non-labeled UFC1. An overlay of representative areas of the [^15^N,^1^H] TROSY-HSQC spectra recorded at 500 MHz are presented. The increasing protein molar ratios are indicated with a rainbow color code from free R1-R2-R3^325–404^ (red) to 8 molar excess of UFC1 (purple). (**C**) Mapping of CSPs induced by UFC1 on the R1-R2-R3^325–404^ sequence. The CSP values (shown as bars) below standard deviation (SD), between 1xSD and 2xSD, and above 2xSD are labeled grey, yellow and red, respectively. The small box shows magnification of CSP diagram for UBA5 residues 325–382. The disappearing resonances within the core R3 sequence are also shown as purple bars; the CSP for the R2 residues around A371 are marked blue. (**D**) NMR titration of the ^13^C,^15^N-labeled R3^381–404W^ peptide with non-labeled UFC1 protein performed at 800 MHz. The same spectral areas as in (**B**) are shown and the same color code is used. (**E**) NMR titration of ^15^N,^13^C-labeled UFC1 with non-labeled R3^381–404W^ peptide recorded at 950 MHz. An overlay of representative areas of the [^15^N,^1^H] TROSY-HSQC spectra is presented. Titration steps are visualized in a rainbow color code. Most significant CSP are highlighted by arrows. Dashed arrows indicate that the initial or final peak position is outside of the presented area.

**Figure 3 ijms-22-07390-f003:**
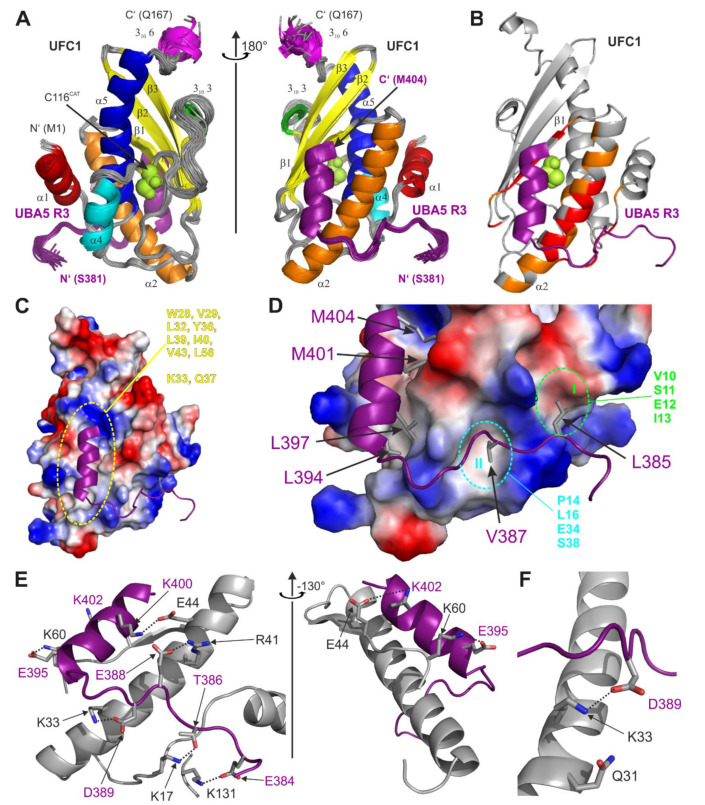
NMR structure of the complex between UFC1 and the UBA5 R3^381–404W^ peptide. (**A**) NMR solution structure of the complex between UFC1 and R3^381–404W^ peptide in two different orientations. All 20 conformers are superimposed over the structured UFC1 core (residues 3–162). All UFC1 secondary structure elements are marked by the following colors: α1—red; α2—orange; 3^10^ helix 3—green; α4—cyan, α5—blue; 3^10^ helix 6—magenta; all β-strands (β1, β2, β3) are yellow. R3^381–404W^ chains are shown in purple. (**B**) Mapping of UFC1 CSPs upon titration with R3^381–404W^ on a representative complex structure (conformer 6, the same orientation as in the A, right plot). The CSP values below standard deviation (SD), between 1xSD and 2xSD, and above 2xSD are labeled grey, yellow and red, respectively. Residues which were not assigned are presented in grey as well. (**C**) UFC1 molecule (conformer 6, the same orientation as in the A, right plot) is shown as a surface with calculated potentials, whereas the R3^381–404W^ molecule is presented by ribbon diagram (purple). The large hydrophobic groove between UFC1 α-helix α2 and β-strand β1 is highlighted with a dashed yellow line. UFC1 residues contributing to the groove formation are listed. (**D**) Hydrophobic patches on UFC1 surface mediating interactions with the UBA5 R3^381–404W^ L385 and V387 side chains are shown as grey sticks. The UFC1 hydrophobic patches I and II are marked with dashed lines (green and magenta, respectively). UFC1 residues forming these patches are listed. (**E**) Polar interactions within the UFC1:R3^381–404W^ complex. Intermolecular hydrogen bonds are shown as dashed lines. (**F**) Detailed view on the intermolecular hydrogen bond between UBA5 D389 and UFC1 K33. The UBA5 Q31 sidechain is also presented as sticks.

**Figure 4 ijms-22-07390-f004:**
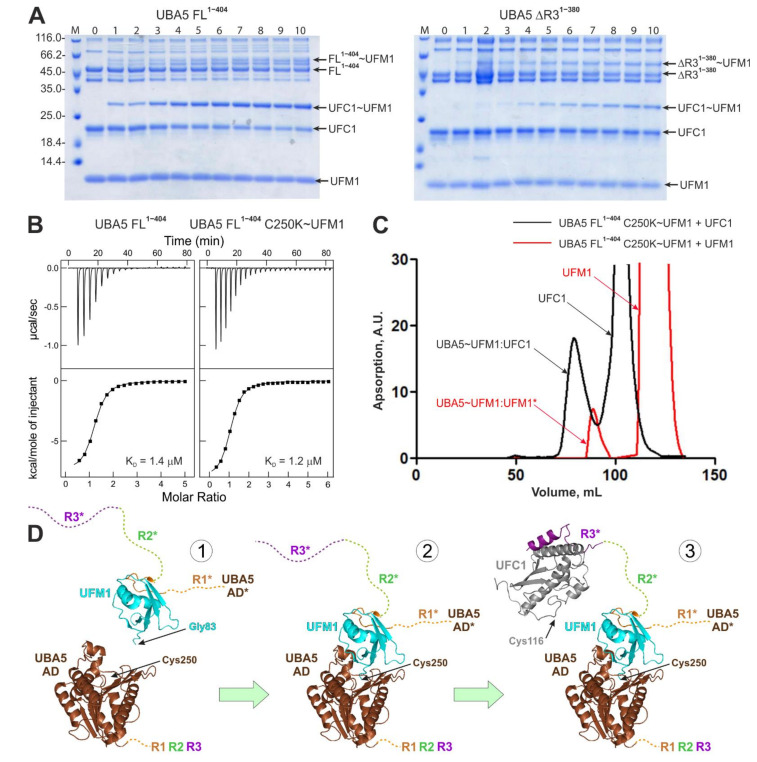
Interaction studies between full length UBA5 and UFC1 proteins. (**A**) Gel electrophoresis of ufmylation assays including UBA5 FL^1–404^ (left plot) and UBA5 ΔR3^1–380^. (**B**) UFC1 binding to full length UBA5 (left plot) and full length stable UBA5~UFM1 conjugate (right plot) observed by ITC experiments. The upper graph displays the raw heat data; the lower graph shows the integrated heat per titration steps (black squares) with best-fit curve (line). K_D_ values are indicated. (**C**) Gel-filtration profiles of the FL^1–404^ C250K~UFM1 conjugates in presence of 4 times molar excess of UFM1 (red lines) and UFC1 (black lines). The peak subjected to electrophoretic analysis is indicated by an asterisk. (**D**) Scheme of reactions involving UBA5 in the ufmylation cascade. The structures of UBA5 AD (brown), UFM1 (cyan) and UFC1 (grey) are represented as ribbon diagrams; the UBA5 unstructured C-terminus containing regions R1 (orange), R2 (green) and R3 (violet) is shown as dashed lines. The structures were generated from PDB entry 5IAA [29]. * indicates regions of another UBA5 molecule involved in the *in trans* transfer of UFM1.

**Table 1 ijms-22-07390-t001:** A list of DNA constructs used in this study.

DNA Construct	Expressed Protein/Peptide	Short Description	References
pET39_Ub19_UBA5^1–404^	FL^1–404^	Full length UBA5, residues 1–404	[31]
pET39_Ub19_UBA5^325–404^	R1-R2-R3^325–404^	UBA5 C-terminal part, residues 325–404	[31]
pET39_Ub19_UFM1	UFM1	Full length UFM1, residues 2–83	[31]
pETm60_Ub3_LC3A	LC3A	LC3A, residues 4–120	[36]
pETm60_Ub3_LC3B	LC3B	LC3B, residues 5–120	[36]
pET39_Ub19_LC3C	LC3C	LC3C, residues 5–126	[36]
pET39_Ub19_GABARAP	GABARAP	GABARAP, residues 3–116	[36]
pETm60_Ub3_GABARAPL1	GABARAPL1	GABARAPL1, residues 2–116	[36]
pET39_Ub19_GABARAPL2	GABARAPL2	GABARAPL2, residues 3–116	[36]
pETm60_Ub	Ubiquitin	Ubiquitin, residues 1–76	[37]
pET39_Ub19_UFC1	UFC1	Full length UFC1, residues 1–167	This work
pET39_Ub19_UBA5^1–330^	AD^1–330^	UBA5 adenylation domain, residues 1–330	This work
pET39_Ub19_UBA5^325–376^	R1-R2^325–376^	UBA5 C-terminal regions R1 and R2, residues 325–376	This work
pET39_Ub19_UBA5^359–404^	R2-R3^359–404^	UBA5 C-terminal regions R2 and R3, residues 359–404	This work
pET39_Ub19_UBA5^325–357^	R1^325–357^	UBA5 C-terminal region R1, residues 325–357	This work
pET39_Ub19_UBA5^359–376^	R2^359–376^	UBA5 C-terminal region R2, residues 359–376	This work
pET39_Ub19_UBA5^388–404^	R3^388–404^	UBA5 C-terminal region R3, residues 388–404	This work
pET39_Ub19_UBA5^381–404W^	R3^381–404W^	Optimized R3, residues 381–404 with C-terminal W	This work
pET39_Ub19_UBA5^325–404^ A371T	R1-R2-R3^325–404^ A371T	UBA5 C-terminal part with A371T mutant (res. 325–404)	This work
pET39_Ub19_UBA5^325–404^ A371E	R1-R2-R3^325–404^ A371E	UBA5 C-terminal part with A371E mutant (res. 325–404)	This work
pET39_Ub19_UBA5^1–380^	ΔR3^1–380^	UBA5 with deleted R3 region, residues 1–380	This work
pNiC-CTH0_UBA5^1–404^ C250K	FL^1–404^ C250K	Full length UBA5 (res. 1–404) with C250K mutant	This work
pET39_Ub19_UBA5^1–330^ C250K ^0^	AD^1–330^ C250K	UBA5 adenylation domain with C250K mutant	This work
pNiC-CTH0_UFC1	UFC1_His6	Full length UFC1 with C-terminal hexahistidine-tag	This work

**Table 2 ijms-22-07390-t002:** Thermodynamic parameters of the interactions between UBA5 C-terminal regions and UBA5-interacting proteins.

Proteins	UBA5 Regions	ΔH (kcal mol^−1^)	ΔS (cal mol^−1^ K^−1^)	−T*ΔS (kcal mol^−1^)	ΔG (kcal mol^−1^)	K_A_*10^−6^ (M^−1^)	K_D_ (µM)	N
UFM1	R1-R2-R3^325–404^	−5.83 ± 0.20 *	4.41	−1.31	−7.15	0.17 ± 0.02	5.8	1.04 ± 0.03
	R1-R2^325–376^	−5.61 ± 0.21	4.99	−1.49	−7.10	0.16 ± 0.01	6.2	0.96 ± 0.03
	R2^359–376^	ND				ND	>100 **	ND
	R1-R2-R3^325–404^ A371T	−10.99 ± 0.25	−13.3	3.96	−7.02	0.14 ± 0.01	7.1	1.12 ± 0.02
	R1-R2-R3^325–404^ A371E	−11.42 ± 0.42	−15.3	4.56	−6.86	0.11 ± 0.01	9.2	1.01 ± 0.01
UFC1	R1-R2-R3^325–404^	−7.04 ± 0.07	2.09	−0.62	−7.66	0.41 ± 0.02	2.4	1.03 ± 0.01
	R2-R3^359–404^	−8.08 ± 0.10	1.59	−0.47	−7.60	0.37 ± 0.01	2.7	0.97 ± 0.01
	R3^388–404^	−4.99 ± 0.22	6.91	−2.06	−7.05	0.03 ± 0.003	10	0.95 ± 0.03
	R2^359–376^	ND				ND	>50 **	ND
	R1-R2-R3^325–404^ A371T	−7.88 ± 0.08	−0.19	0.06	−7.82	0.54 ± 0.03	1.8	1.03 ± 0.008
	R1-R2-R3^325–404^ A371E	−7.78 ± 0.05	0.36	−0.11	−7.88	0.60 ± 0.02	1.6	1.03 ± 0.005
	FL^1–404^	−7.62 ± 0.01	1.26	−0.38	−8.00	0.72 ± 0.04	1.4	0.97 ± 0.009
	FL^1–404^ C250K~Ufm1	−8.21 ± 0.01	−0.34	0.10	−8.10	0.87 ± 0.05	1.2	1.03 ± 0.009
GABARAPL2	R1-R2-R3^325–404^	−8.64 ± 0.06	2.04	−0.61	−9.25	5.99 ± 0.49	0.17	0.97 ± 0.004
	R1-R2^325–376^	−8.08 ± 0.05	4.44	−1.32	−9.41	7.87 ± 0.74	0.13	0.91 ± 0.003
	R2^359–376^	ND				ND	>100 **	ND
	R1-R2-R3^325–404^ A371T	−7.58 ± 0.07	7.76	−2.31	−9.89	17.90 ± 3.75	0.06	0.937 ± 0.005
	R1-R2-R3^325–404^ A371E	−7.79 ± 0.05	7.23	−2.16	−9.95	19.60 ± 2.57	0.06	1.01 ± 0.003
GABARAP	R1-R2-R3^325–404^	−0.93 ± 0.04	24.2	−7.22	−8.15	0.96 ± 0.16	1.1	0.99 ± 0.03
	R1-R2^325–376^	−1.1 ± 0.02	23.1	−6.89	−7.99	0.72 ± 0.05	1.4	0.94 ± 0.01
LC3B	R1-R2-R3^325–404^	−4.47 ± 0.10	8.86	−2.64	−7.33	0.24 ± 0.09	4.2	0.92 ± 0.02
	R1-R2^325–376^	−4.23 ± 0.09	10.7	−3.19	−7.42	0.27 ± 0.14	3.7	0.98 ± 0.02
	R2^359–376^	ND				ND	>100 *	ND
	R1-R2-R3^325–404^ A371T	−3.76 ± 0.05	14.3	−4.26	−8.02	0.76 ± 0.04	1.3	0.937 ± 0.009
	R1-R2-R3^325–404^ A371E	−3.93 ± 0.05	15.3	−4.56	−8.49	1.66 ± 0.12	0.6	0.944 ± 0.009
LC3A	R1-R2-R3^325–404^	4.25 ± 10.5	10.5	−3.13	−7.38	0.26 ± 0.03	3.8	0.91 ± 0.04
	R1-R2^325–376^	−3.81 ± 0.18	11.6	−3.46	−7.26	0.21 ± 0.02	4.7	0.94 ± 0.03
UBA5 AD^1–330^	R1-R2-R3^325–404^	ND				ND	-	ND
Ub	R1-R2-R3^325–404^	ND				ND	-	ND

* Here and further the ± sign corresponds to a fitting error of the individual experiment. ** Estimated value.

## Data Availability

The structure of UFC1:UBA5 R3 complex was deposited in the Protein Data Bank (https://www.rcsb.org/) (accessed on 7 July 2021) with accession code 7OVC. The chemical shift assignments were deposited in the BioMagResBank database (https://bmrb.io) (accessed on 7 July 2021) with accession code 34638.

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
