# Peer review of "A Concerted Action of UBA5 C-Terminal Unstructured Regions Is Important for Transfer of Activated UFM1 to UFC1"

_ijms, 2021, doi:10.3390/ijms22147390_

Round 1
Reviewer 1 Report
The manuscript explores the interaction profile between the UBA5 and UFC1 enzymes by through binding studies and solving the structure of a UBA5 peptide and the UFC1. This report is very thorough, organized, and detailed, and should serve to advance the field. I have only a few minor comments:
For the weaker binding interactions (Figure 2A; Supplementary Figure S2), is it possible to use higher concentrations in the ITC to measure more accurate Kd values?
In Table 2, is the error fitting error? Or measurement error? How many times was each ITC measurement repeated?
In your Methods section, please give more experimental details for your ITC measurements. e.g., What was the stirring speed? How did you account for the heats of dilution?
Some of your figures have fonts that are very small and are very difficult to read. Please double-check font size in all figures. In particular, all superscripts are difficult to decipher in Figure 1, and the specific constructs used are important to the storyline of your data. Another example is Supplementary Figure S2, please make this figure much larger. Third, in supplementary Figure S2, the NMR assignment labels are difficult to read. Maybe only highlight a few representative residues in larger font? Or provide a much larger plot?
Author Response
Reviewer comments:
We thank all the reviewers for the positive feedback, it is important for us to know that the independent referees found our work of interest for other researchers. Indeed, the reviewer’s comments and suggestions were very fair and permitted us to improve the quality of our Manuscript. We respond to all the questions raised by reviewers, and we hope that we could address all of them. Please find below our point-by-point responses.
Reviewer #1:
The manuscript explores the interaction profile between the UBA5 and UFC1 enzymes by through binding studies and solving the structure of a UBA5 peptide and the UFC1. This report is very thorough, organized, and detailed, and should serve to advance the field.
Reply: Thanks this reviewer for the kind estimation of our work!
I have only a few minor comments:
- For the weaker binding interactions (Figure 2A; Supplementary Figure S2), is it possible to use higher concentrations in the ITC to measure more accurate Kd values?
Reply: Thanks reviewer for this comment. In fact, the concentrations of UBA5 C-terminal peptides used for the ITC experiments were experimentally optimized to avoid aggregation effects (started from 1 mM concentration). Therefore, one cannot increase significantly the concentration of UBA5 C-terminal peptides without increasing its uncertainty. In other words, increasing of the peptides concentrations should be more than 20 times to bring the constant C, which describes the practical window of the instrument to accurately determine the binding constants (first evaluated in Wiseman et al., 1989), to a reasonable value 5. It corresponds to the 8 mM effective peptide concentration in syringe, which is impossible to prepare due to aggregation problems as written above.
- In Table 2, is the error fitting error? Or measurement error? How many times was each ITC measurement repeated?
Reply: We have performed a single ITC experiment for each individual type of interactions, which is acceptable for such experimental set. Therefore, all the error values in Table 2 correspond to the fitting error. We indicated that in the Table 2 subscriptions and in the Material and Method section (4.4 Isothermal Titration Calorimetry).
- In your Methods section, please give more experimental details for your ITC measurements. e.g., What was the stirring speed? How did you account for the heats of dilution?
Reply: To obtain a dilution heat in each ITC experiment, we performed additional titration of the UBA5 C-terminal peptides (in syringe) into the ITC buffer (in measuring cell). Dilution heats of the proteins/peptides in measuring cell were neglected as their heat effect was equal to the heat effect of buffer-to-buffer titrations. The stirring speed was set to 307 rpm, initial delay to 180 seconds, and spacer between the titration steeps to 200 seconds. The later was estimated experimentally to avoid a kinetic contribution to the observed heat effects. We added all the experimental details to the part 4.4 Isothermal Titration Calorimetry in the Material and Method section.
- Some of your figures have fonts that are very small and are very difficult to read. Please double-check font size in all figures. In particular, all superscripts are difficult to decipher in Figure 1, and the specific constructs used are important to the storyline of your data. Another example is Supplementary Figure S2, please make this figure much larger. Third, in supplementary Figure S3, the NMR assignment labels are difficult to read. Maybe only highlight a few representative residues in larger font? Or provide a much larger plot?
Reply: We thank reviewer for this comment. In order to achieve a good reading, we increased fonts size and a resolution of all figures. Additionally, we have modified assignments labels according to the reviewer’s suggestions and also increase the resolution.
Reviewer 2 Report
comments attached as a pdf file

Author Response
Reviewer comments:
We thank all the reviewers for the positive feedback, it is important for us to know that the independent referees found our work of interest for other researchers. Indeed, the reviewer’s comments and suggestions were very fair and permitted us to improve the quality of our Manuscript. We respond to all the questions raised by reviewers, and we hope that we could address all of them. Please find below our point-by-point responses.
Reviewer #2:
In this manuscript, the authors presented biochemical and biophysical focused on defining the functionally interactions between UBA5, an E1 enzyme, and UFC1, an E2 enzyme, involved in the ufmylation cascade. The authors combined ITC and NMR techniques with biochemical characterizations. The main findings include identification of a C-terminal region of UAF, termed “R3”, which strongly interact with UFC1. The authors solved the solution NMR structure of the peptide corresponding to “R3” bound to UFC1. Biochemical assays showed that isolated AD and R1-R2-R3 are sufficient to transfer activated UFM1.
Overall, the manuscript is very well written. The conclusions are all supported by data, which are clearly presented. I suggest publication of this manuscript after minor revision.
Reply: Thanks this reviewer for the kind estimation of our work!
- The formula used to calculate chemical shift perturbations (CSPs) should be given. In Figure 2C, the actual CSPs should be given for resonances that display higher than 2 X SD CSPs. In current format, the Y-axis is truncated at ~ 2 X SD. It is not clear to me why this was done. Also, some bars are purple, which is not explained in the figure caption. The authors should use different colors to distinguish between resonance that show large CSPs (> 2 x SD) and those that disappear due to intermediate exchange, rather than using the red color for both.
Reply: We thank reviewer for this comment. We indicated the formula for CSP calculation in the part 4.5 NMR spectroscopy in the Material and Method section:
“CSP values, Δδ, were calculated for each individual amide group using the formula: Δδ=[(ΔδN/5)2 + Δδ2HN)]1/2.”
We also modified the Figure 2C according to the reviewer suggestion and described color-code for CSPs in the figure legends precisely.
- In Table 2, “KA*106 [M-1]” should be replaced by “KA [106 M-1]” because the numbers are not KA * 106; they are actually KA * 10-6
Reply: Correct. We modified the Table 2 according to this comment indicating that all presented KA values are 10-6 from the real KA values. We would not change the dimension of the KA values (M-1).
- In line 551, in “NOESY-[13Cali,1H}-HSQC”, the curly brace should be replaced by “]”.
Reply: Correct. We modified the line 566 (former 551) to read:
“strains, 3D F1-13C/15N-filtered NOESY-[13Cali,1H]-HSQC, NOESY-[13Caro,1H]-SOFAST-“
- The full names of many NMR experiments are not given under abbreviations, e.g. HMQC, NOESY, TOCSY, BEST-TROSY etc.
Reply: Thanks reviewer for indicating that. We have added all the NMR abbreviations (usually accepted in specialized journals as a “textbook-knowledge”) to the Abbreviations section for broader audience.